# Synergy between loss of *NF1* and overexpression of *MYCN* in neuroblastoma is mediated by the GAP-related domain

Shuning He[1]*, Marc R Mansour[1,2], Mark W Zimmerman[1], Dong Hyuk Ki[1], Hillary M Layden[1], Koshi Akahane[1], Evisa Gjini[3], Eric D de Groh[4,5,6], Antonio R Perez-Atayde[7], Shizhen Zhu[1,8,9], Jonathan A Epstein[4,5,6], A Thomas Look[1]*

[1]Department of Pediatric Oncology, Dana-Farber Cancer Institute, Harvard Medical School, Boston, United States; [2]Department of Hematology, UCL Cancer Institute, University College London, London, United Kingdom; [3]Center for Immuno-Oncology, Dana-Farber Cancer Institute, Harvard Medical School, Boston, United States; [4]Department of Cell and Developmental Biology, Perelman School of Medicine at the University of Pennsylvania, Philadelphia, United States; [5]Penn Cardiovascular Institute, Perelman School of Medicine at the University of Pennsylvania, Philadelphia, United States; [6]Institute for Regenerative Medicine, Perelman School of Medicine at the University of Pennsylvania, Philadelphia, United States; [7]Department of Pathology, Children's Hospital Boston, Harvard Medical School, Boston, United States; [8]Department of Biochemistry and Molecular Biology, Mayo Clinic College of Medicine, Rochester, United States; [9]Department of Molecular Pharmacology and Experimental Therapeutics, Mayo Clinic College of Medicine, Rochester, United States

*For correspondence: shuning_he@dfci.harvard.edu (SH); thomas_look@dfci.harvard.edu (ATL)

Competing interests: The authors declare that no competing interests exist.

**Abstract** Earlier reports showed that hyperplasia of sympathoadrenal cell precursors during embryogenesis in *Nf1*-deficient mice is independent of Nf1's role in down-modulating RAS-MAPK signaling. We demonstrate in zebrafish that *nf1* loss leads to aberrant activation of RAS signaling in *MYCN*-induced neuroblastomas that arise in these precursors, and that the GTPase-activating protein (GAP)-related domain (GRD) is sufficient to suppress the acceleration of neuroblastoma in *nf1*-deficient fish, but not the hypertrophy of sympathoadrenal cells in *nf1* mutant embryos. Thus, even though neuroblastoma is a classical "developmental tumor", NF1 relies on a very different mechanism to suppress malignant transformation than it does to modulate normal neural crest cell growth. We also show marked synergy in tumor cell killing between MEK inhibitors (trametinib) and retinoids (isotretinoin) in primary *nf1a-/-* zebrafish neuroblastomas. Thus, our model system has considerable translational potential for investigating new strategies to improve the treatment of very high-risk neuroblastomas with aberrant RAS-MAPK activation.

## Introduction

Neuroblastoma, a malignant embryonic tumor of childhood, arises in neural crest-derived dopaminergic neuroblasts that generate the peripheral sympathetic nervous system (PSNS). This cancer is the most common noncranial solid tumor in childhood, accounting for 8–10% of all childhood

**eLife digest** Neuroblastoma is one of the most common childhood cancers and is responsible for about 15% of childhood deaths due to cancer. The neuroblastoma tumors arise in cells that develop into and form part of the body's nervous system. Many researchers have studied the genetics of this disease and have recognised common patterns. In particular, neuroblastomas can occur when a protein called MYCN is over-produced and a tumor suppressor protein called NF1 is lost.

NF1 is a large protein with several distinct parts or domains. The most studied domain of NF1 is called the GRD, and it is mainly responsible for dampening down signals that cause cells to grow, specialize and survive. However, experiments in mice have revealed that this protein uses its other domains to control the normal development of part of the nervous system.

He et al. wanted to know which domains of NF1 are important for suppressing the growth of neuroblastomas. The experiments were conducted in zebrafish that had been engineered to produce an excess of the human version of MYCN. When He et al. also deleted the gene for the zebrafish's version of NF1, the fish quickly developed neuroblastomas. Supplying the zebrafish with just the GRD of NF1 was enough to supress the growth of the tumors. These experiments show that NF1 uses different domains and signalling pathways to regulate the normal development of part of the nervous system and to prevent formation of neuroblastoma.

These engineered zebrafish represent an animal model of neuroblastoma that mimics the human disease in many ways. This model will make it possible to test new drug combinations and to find more effective treatments for neuroblastoma patients.

malignancies and 15% of all cancer deaths in children (*Maris and Matthay, 1999*). Neuroblastoma is genetically heterogeneous, with multiple interacting genetic mutations required to generate fully transformed malignant tumors. As in many other pediatric cancers, resequencing studies have documented a very low frequency of somatic mutations in neuroblastoma (*Cheung et al., 2012*; *Molenaar et al., 2012*; *Pugh et al., 2013*), including *ALK* (~9.2% of cases), *PTPN11* (~2.9%), *ATRX* (~2.5%) and *NF1* (~1%). Amplification and overexpression of the *MYCN* oncogene occurs in about one-third of patients and is a principal indicator of a poor prognosis (*Brodeur et al., 1984*; *Hansford et al., 2004*). Genomic aberrations that inactivate the *NF1* tumor suppressor gene, including loss-of-function mutations and deletions, as well as decreased expression levels of the gene, have been identified in 6% of primary neuroblastomas and are predictive of a poor outcome (*Hölzel et al., 2010*), suggesting an important role for *NF1* loss in neuroblastoma tumorigenesis.

The *NF1* gene encodes neurofibromin, a 2818 amino acid protein whose main functional domain is the ~330 amino acid GTPase-activating protein-related domain (GRD), which negatively regulates RAS signaling by catalyzing the hydrolysis of RAS-GTP into RAS-GDP (*Nur-E-Kamal et al., 1993*); thus, one consequence of *NF1* loss is the aberrant activation of RAS signaling (*Maertens and Cichowski, 2014*). Loss of *NF1* in neuroblastoma cells has been shown to mediate resistance to retinoic acid via hyperactive RAS signaling, which can be abolished by enforced expression of *NF1*-GRD (*Hölzel et al., 2010*). However, it is known that NF1 has other functions in PSNS development besides the downmodulation of RAS signaling, because *Nf1* mutant mice die at birth with evidence of massive overgrowth of neural crest tissues, including the sympathetic ganglia, while overexpression of the GRD domain is unable to reverse this overgrowth (*Ismat et al., 2006*). In addition, studies showed identical or only modestly elevated RAS-GTP levels in *NF1*-deficient human neuroblastoma cells, in contrast to highly elevated RAS-GTP levels in *NF1*-deficient Schwannoma tumor cells (*Johnson et al., 1993*; *The et al., 1993*). These results, coupled with the numerous mutations of *NF1* that cause the disease neurofibromatosis type 1, but do not appear to affect protein stability or GAP function (*Abernathy et al., 1997*; *Fahsold et al., 2000*), argue that functional domains outside the GRD may mediate important aspects of neurofibromin function in neuroblastoma tumor suppression.

In earlier work, we identified two separate duplicated *nf1* zebrafish genes, *nf1a* and *nf1b*, and generated multiple loss-of-function *nf1* mutant zebrafish lines affecting both of these alleles

(*Lee et al., 2010*; *Padmanabhan et al., 2009*; *Shin et al., 2012*). Mutant larvae carrying at least one wild-type *nf1a* or *nf1b* allele are viable, fertile, and show no obvious phenotypes during early development. To gain insight into the cellular and molecular consequences of *NF1* loss in neuroblastoma, we used transgenic zebrafish models of neuroblastoma that overexpresses human MYCN in the PSNS (*Zhu et al., 2012*).

Here, we report that loss of the *nf1a* orthologue, which is much more highly expressed than *nf1b* during early PSNS development, greatly accelerates the onset of neuroblastoma induced by *MYCN* overexpression, with nearly complete penetrance by 5 weeks of age in *nf1*-deficient zebrafish. Loss of *nf1* led to the aberrant activation of RAS signaling in MYCN-induced neuroblastoma, promoting both tumor cell survival and proliferation. We also show that the very aggressive growth properties of MYCN-induced neuroblastomas with loss of *nf1* are due to aberrant activation of RAS signaling, because the increased penetrance and rapid growth could be suppressed by overexpressing the intact NF1 GRD domain. These findings establish *nf1*-deficient zebrafish that overexpress *MYCN* as an ideal animal model system for investigating new strategies to improve treatment of very high-risk neuroblastomas with aberrant RAS-MAPK activation.

In vivo structure-function analysis with both the wild-type and inactive GRD domain of *NF1* revealed that the GAP activity of the GRD domain is required for the tumor suppressor function of NF1 in neuroblastoma. By contrast, the wild-type GRD domain failed to rescue the hypertrophy of sympathoadrenal cells in *nf1* mutant embryos, indicating that the role of NF1 in suppressing neuroblastoma tumorigenesis differs from the mechanism that prevents PSNS hyperplasia during normal development.

## Results

### *nf1* restricts PSNS cell growth during normal embryologic development independent of the GAP activity of GRD domain

Forced expression of the NF1 GTPase-activating protein-related domain (GRD) has been used to restore GAP activity in *NF1*-deficient human and mouse cells (*Hölzel et al., 2010*; *Ismat et al., 2006*; *Keutmann et al., 1983*); however, this domain did not rescue the developmental overgrowth of neural crest-derived PSNS tissues that is observed in *Nf1*-deficient mice (*Brannan et al., 1994*; *Gitler et al., 2003*; *Ismat et al., 2006*), supporting an alternative activity of Nf1 as the mediator of growth regulation in PSNS neuronal progenitors. For our studies in the zebrafish model, we began with experiments to confirm this surprising result in zebrafish PSNS precursor cells by studying the effect of loss of *nf1* on growth of the superior cervical ganglia (SCG) during the normal development of early embryos (*Figure 1*). We bred the *nf1a+/-; nf1b+/-* mutant zebrafish line (*Shin et al., 2012*) with transgenic fish overexpressing either EGFP or mCherry in the PSNS under control of the *dβh* promoter (*dbh:EGFP* or *dbh:mCherry*) (*Zhu et al., 2012*). As in the mouse, complete loss of *nf1* led to increased cell numbers in the SCG at 6 dpf (compare *Figure 1D* with panel A, also panel E). Homozygous loss of *nf1a* led to the same level of increase in SCG neuronal cell number as homozygous loss of *nf1a* plus *nf1b*, while the loss of *nf1b* had little effect on SCG cell numbers (*Figure 1B and G*), which is consistent with the fact that *nf1a* is expressed at a much higher level than *nf1b* in sympathetic neurons as well as the whole embryo during the first week of zebrafish embryonic development (*Figure 1F*). Later in development at 3, 4 and 6 weeks of life, we observed lower relative *nf1a* and *nf1b* levels in RNA from the whole fish, and at these time points the expression levels of *nf1a* and *nf1b* were similar to each other, without evidence of the predominance of *nf1a* that was observed at 1 week of age.

Given the high conservation between the GRD domains of the human and zebrafish neurofibromin proteins (*Padmanabhan et al., 2009*), we introduced the GRD domain of human NF1 into our *nf1* mutant zebrafish using a *dbh:GRD; dbh:mCherry* stable transgenic zebrafish line that expresses both this domain and the *dbh:mCherry* fluorescent marker in the PSNS. In agreement with studies in the mouse, we found that the functionally active wild-type GRD domain (designated 'wt-GRD') did not rescue SCG overgrowth in *nf1* mutant fish, as the levels of SCG overgrowth in *nf1a-/-; nf1b+/+; EGFP* embryos transgenic for *dbh:wt-GRD;dbh:mCherry* were similar to those of *nf1a-/-; nf1b+/+; EGFP* embryos at 6 dpf (*Figure 1H*). Thus, nf1 is required to control the growth of PSNS precursor

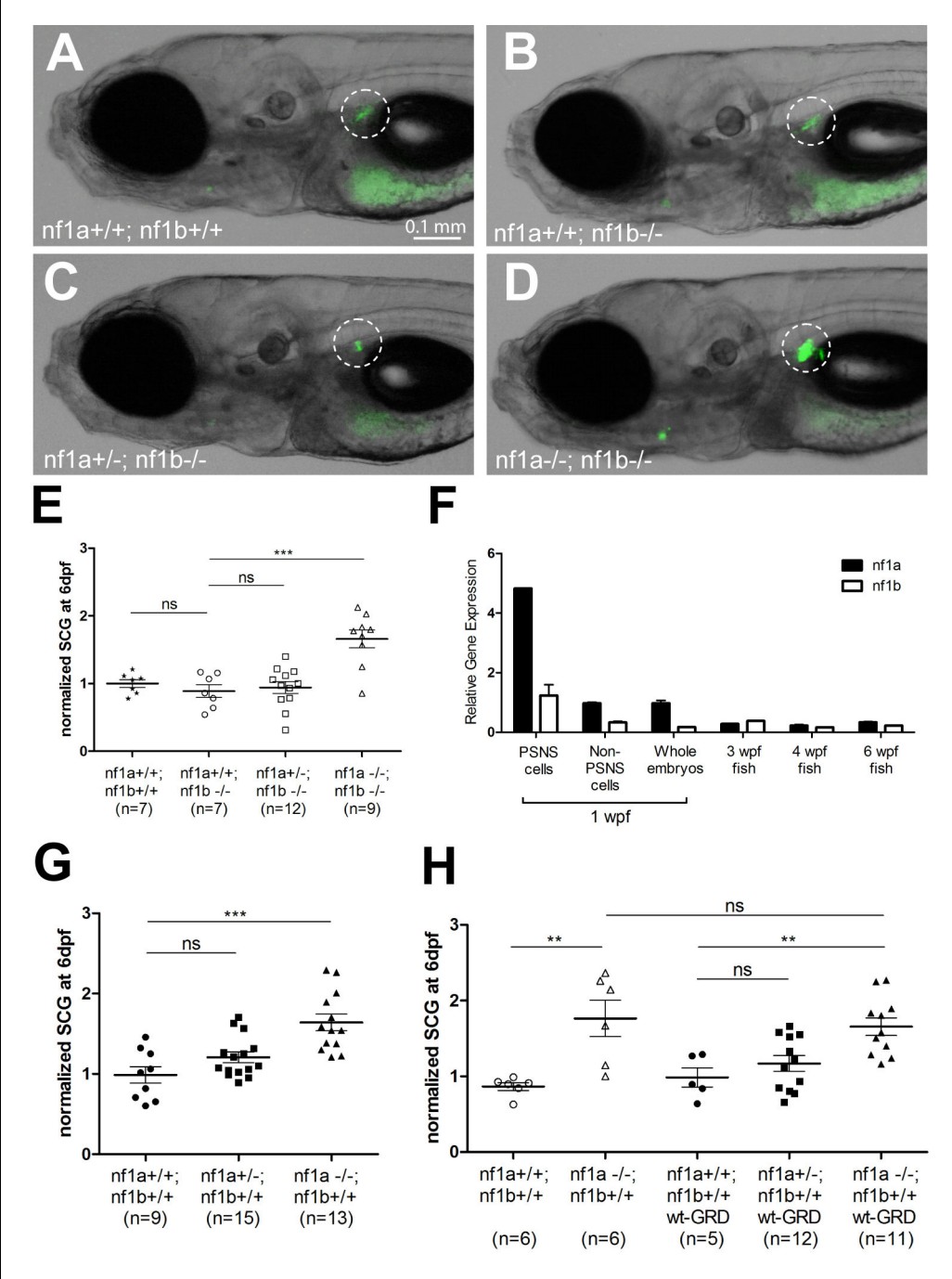

**Figure 1.** The GRD domain of NF1 cannot rescue the PSNS overgrowth in *nf1* deficient zebrafish. (**A–D**) Development of superior cervical ganglia (SCG, highlighted by dotted circles) in representative embryos of *nf1a+/ +;nf1b+/+;GFP, nf1a+/+;nf1b-/-;GFP, nf1a+/-;nf1b-/-;GFP* and *nf1a-/-;nf1b-/-;GFP* genotypes at the age of 6 days postfertilization (dpf). (**E**) Quantification of GFP+ cells in the SCG of embryos of *nf1a+/+;nf1b+/+;GFP* (n = 7), *nf1a +/+;nf1b-/-;GFP* (n = 7), *nf1a+/-;nf1b-/-;GFP* (n = 12) and *nf1a-/-;nf1b-/-;GFP* (n = 9) genotypes at the age of 6 dpf. (**F**) Quantitative RT-PCR showing the relative expression levels of *nf1a* and *nf1b* in the PSNS of zebrafish embryos and juveniles. 1-week-old *dbh:mCherry* embryos (n = 200) were pooled, dissociated and FACS sorted to obtain PSNS (mCherry+) and non-PSNS (mCherry-) cells for analysis. RNA of whole embryos from the same clutch of eggs were also analyzed. RNAs from whole juvenile zebrafish at the ages of 3, 4, and 6 weeks were also examined by quantitative RT-PCR. (**G**) Quantification of GFP+ cells in the SCG of embryos of *nf1a+/+;nf1b+/+; GFP* (n = 9), *nf1a+/-;nf1b+/+;GFP* (n = 15) and *nf1a-/-;nf1b+/+;GFP* (n = 13) genotypes at the age of 6 dpf. (**H**)

*Figure 1 continued on next page*

*Figure 1 continued*

Quantification of GFP+ cells in the SCG of embryos of *nf1a+/+;nf1b+/+;GFP* (n = 6), *nf1a-/-;nf1b+/+;GFP* (n = 6), *nf1a+/+;nf1b+/+;GFP;wt-GRD;mCherry* (n = 5), *nf1a+/-;nf1b+/+;GFP;wt-GRD;mCherry* (n = 12) and *nf1a-/-;nf1b+/+;GFP;wt-GRD;mCherry* (n = 11) genotypes at the age of 6 dpf. **p<0.01, ***p<0.001 by two-tailed unpaired t-test.

cells during embryologic development in zebrafish as well as mice, and this activity is independent of the nf1 GRD in zebrafish as well as in mice.

## Loss of *nf1* accelerates MYCN-induced neuroblastoma in vivo

Loss of *Nf1* has been shown to increase the penetrance of MYCN-induced neuroblastoma in *MYCN* transgenic mice (*Weiss et al., 1997*), and for our studies in zebrafish we determined the extent to which loss of *nf1* synergizes with overexpression of the *MYCN* oncogene in this disease. Thus, we bred the *nf1a+/-; nf1b+/-* mutant zebrafish line (*Shin et al., 2012*) with transgenic fish overexpressing *EGFP-MYCN* in the PSNS under control of the *dβh* promoter (designated 'MYCN' in this article) (*Zhu et al., 2012*). The *nf1a+/-;nf1b+/-;MYCN* zebrafish line was then bred with a *nf1a+/-;nf1b+/-; dbh:EGFP* transgenic line to obtain transgenic lines harboring one, two, or three *nf1* mutant alleles, as well as the overexpression *MYCN* transgene in the PSNS (*Figure 2figure supplement 1*). The resultant offspring were monitored every 2 weeks, beginning at 4 weeks postfertilization for evidence of fluorescent tumors in the PSNS. The fish were genotyped at 8 weeks of age, with each of the 16 expected genotypes represented in the offspring of this cross: (1) *nf1a+/+;nf1b+/+;EGFP*; (2) *nf1a+/-;nf1b+/+;EGFP*; (3) *nf1a+/+;nf1b+/-;EGFP*; (4) *nf1a-/-;nf1b+/+;EGFP*; (5) *nf1a+/-;nf1b+/-; EGFP*; (6) *nf1a+/+;nf1b-/-;EGFP*; (7) *nf1a-/-;nf1b+/-;EGFP*; (8) *nf1a+/-;nf1b-/-;EGFP*; (9) *nf1a+/+;nf1b +/+;EGFP;MYCN*; (10) *nf1a+/-;nf1b+/+;EGFP;MYCN*; (11) *nf1a+/+;nf1b+/-;EGFP;MYCN*; (12) *nf1a-/- ;nf1b+/+;EGFP;MYCN*; (13) *nf1a+/-;nf1b+/-;EGFP;MYCN*; (14) *nf1a+/+;nf1b-/-;EGFP;MYCN*; (15) *nf1a-/-;nf1b+/-;EGFP;MYCN*; (16) *nf1a+/-;nf1b-/-;EGFP;MYCN*. Larval fish harboring homozygous loss of both alleles of *nf1a* and *nf1b* were not obtained from this cross, because they die between 7 and 9 days of age (*Shin et al., 2012*).

Transgenic fish lines that expressed *EGFP* in the PSNS but not the *MYCN* transgene did not develop neuroblastoma, regardless of the mutational status of *nf1*, indicating that loss of up to three alleles of *nf1* was insufficient to induce neuroblastoma on its own (*Figure 2A* and *Figure 4C*). Over the 20-week course of this experiment, neuroblastoma developed in one of the 14 transgenic *MYCN*-positive zebrafish with wild-type *nf1* alleles (*Figure 2C*), consistent with the relatively low penetrance and late onset of these tumors in the wild-type background (*Zhu et al., 2012*). Histopathologically, the tumor masses that developed in *MYCN* transgenic fish with *nf1* mutations consisted of small, undifferentiated tumor cells with distinct single nucleoli (*Figure 2B*), which were diagnostic of neuroblastoma and indistinguishable from the tumors that develop in *MYCN* transgenic zebrafish with wild-type *nf1 alleles* (*Zhu et al., 2012*). The first tumors arose in the interrenal gland (IRG) of 4-week-old animals with loss of one or more of the *nf1a* alleles, representing the earliest onset we have observed in this *MYCN* transgenic model (*Figure 2C*).

Tumor penetrance in the progeny of this cross varied widely depending on the *nf1* genotype (*Figure 2C*). Three genotypes that retained two copies of functional *nf1a* were associated with similar tumor onsets, as shown in blue in *Figure 2C*, with a neuroblastoma penetrance of ~5.5%, suggesting that loss of *nf1b* has much less impact than *nf1a* on the pathogenesis of neuroblastoma in the PSNS. This is consistent with the fact that *nf1b* is expressed at a much lower level than *nf1a* in the PSNS during the first week of zebrafish development (*Figure 1F*). By contrast, the highest neuroblastoma penetrance levels at 4 weeks of age (82.6%) was observed in *MYCN* transgenic fish with homozygous loss of *nf1a* and heterozygous loss of *nf1b*, which produced the earliest onset and highest penetrance of neuroblastoma that we have observed in this model system. Finally, the penetrance of *nf1a-/-* fish with wild-type *nf1b* was still very high at 62.5%, indicating that of the two paralogues, the *nf1a* gene is primarily responsible for tumor suppression in the developing zebrafish sympathetic nervous system.

To evaluate the histopathology of neuroblastoma tumors derived from PSNS cells with loss of *nf1*, we focused on the genotypes with mutant *nf1a* alleles and wild-type *nf1b* alleles.

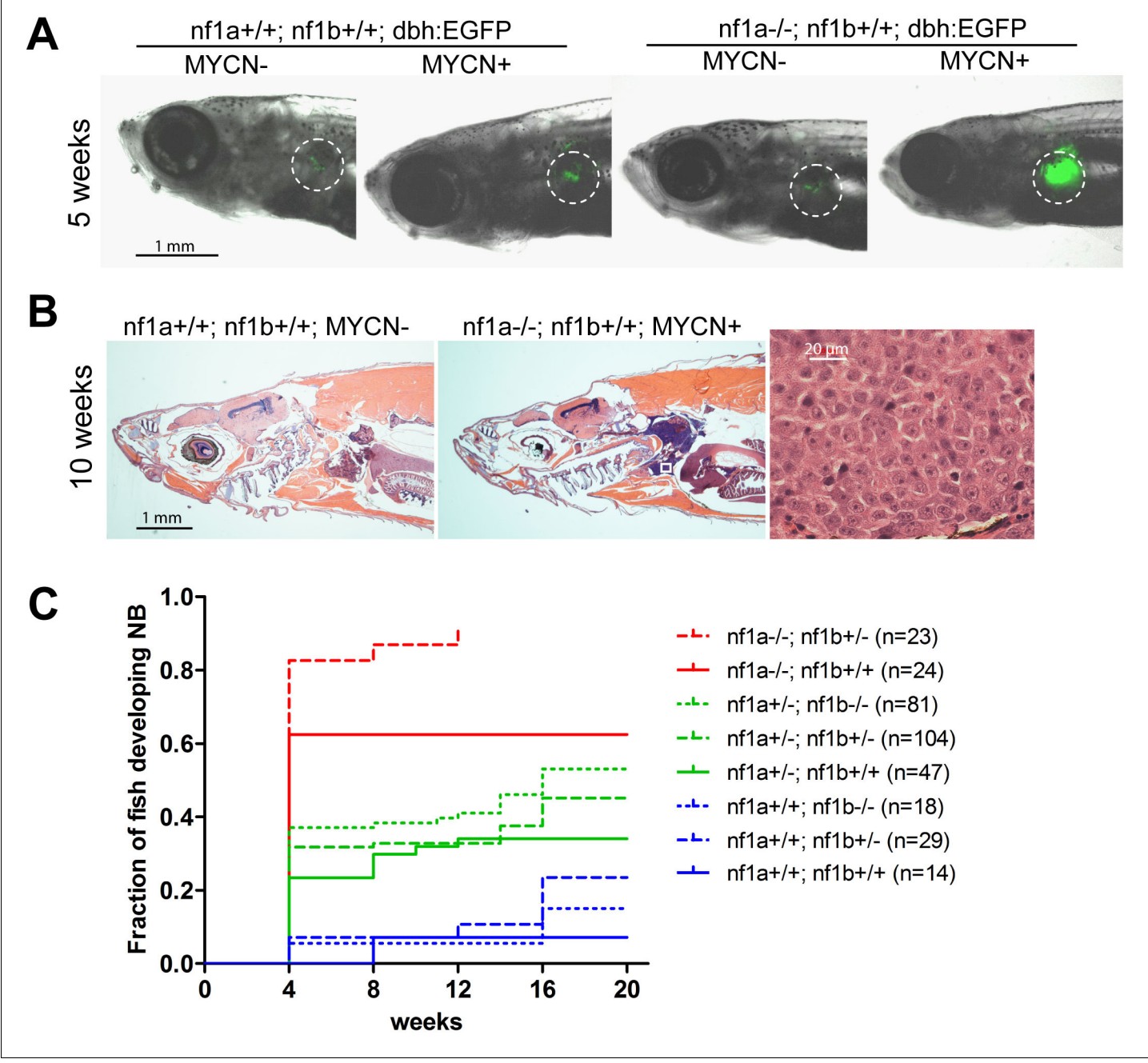

**Figure 2.** Loss of *nf1* accelerates disease onset and increases the penetrance of MYCN-induced neuroblastoma. (**A**) Representative fish of *nf1a+/+;nf1b+/+;GFP, nf1a+/+;nf1b+/+;MYCN;GFP, nf1a-/-;nf1b+/+;GFP* and *nf1a-/-;nf1b+/+;MYCN;GFP* genotypes at 5 weeks of age.The interrenal glands (IRGs) are highlighted with dashed circles. (**B**) H&E-stained sagittal sections of a 10-week old *nf1a+/+;nf1b+/+;GFP* fish (left) and a 10-week old *nf1a-/-;nf1b+/-;GFP;MYCN* fish (middle) and 63-fold magnified tumor cells (right), which are magnified from the area in the small white box in the middle panel. (**C**) Cumulative frequency of neuroblastoma in *MYCN* transgenic zebrafish representing all eight *nf1* genotypes generated by the breeding of the *nf1a+/-;nf1b+/-;MYCN* line with the *nf1a+/-;nf1b+/-* zebrafish line (*p<0.0001 *nf1a-/-;nf1b+/+* vs. *nf1a+/+;nf1b+/+*).

The following figure supplement is available for figure 2:

**Figure supplement 1.** Breeding scheme to obtain zebrafish lines harboring mutated nf1 and overexpression of GFP and the MYCN transgene in the PSNS for this study.

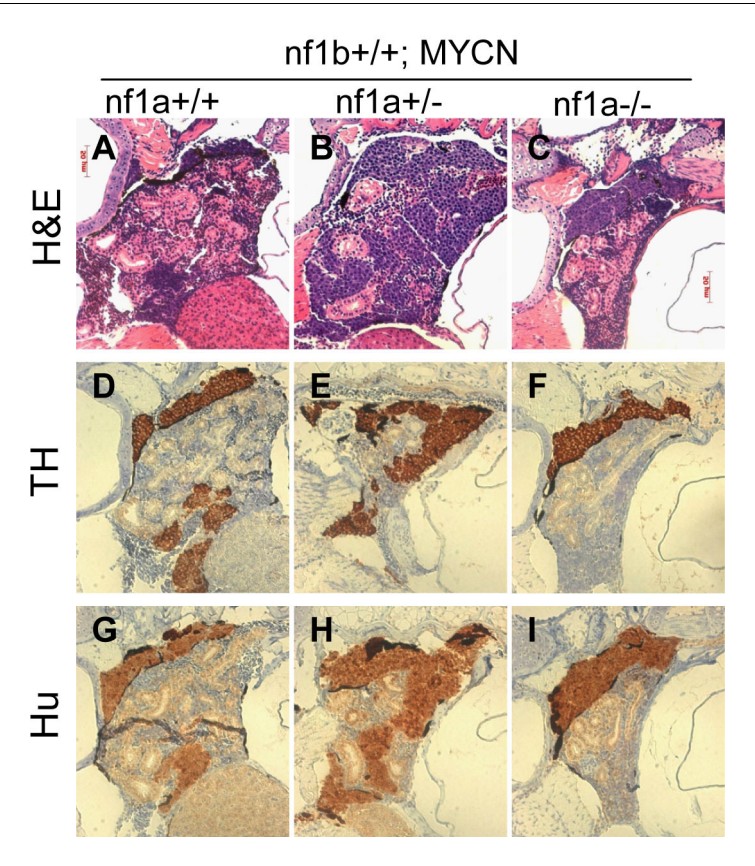

**Figure 3.** Neuroblastomas arise from sympathetic neuroblast precursors in *MYCN* transgenic fish with loss of *nf1*. H&E staining of sagittal sections through the tumors in the IRG of *nf1a+/+;nf1b+/+;MYCN;GFP* (**A**), *nf1a+/-;nf1b+/+;MYCN;GFP* (**B**) and *nf1a-/-;nf1b+/+;MYCN;GFP* (**C**) fish at the age of 6 weeks. Immunohistochemical analysis of neuroblastoma markers tyrosine hydroxylase (TH, **D–F**) and Hu (**G–I**) expression on sagittal sections through tumors in the IRG of *nf1a+/+;nf1b+/+;MYCN;GFP* (**D,G**), *nf1a+/-;nf1b+/+;MYCN;GFP* (**E,H**) and *nf1a-/-;nf1b+/+;MYCN;GFP* (**F,I**) fish at the age of 6 weeks.

The following figure supplement is available for figure 3:

**Figure supplement 1.** No TH+, Hu+ neuroblasts was detected in the IRG of nf1a mutant zebrafish which had no overexpression of MYCN.

Neuroblastomas that arose in fish with *nf1a+/-;nf1b+/+;EGFP;MYCN* and *nf1a-/-;nf1b+/+;EGFP;MYCN* genotypes were all strongly immunoreactive for tyrosine hydroxylase (TH) and the pan-neuronal marker Hu, comparable to the fish with wild-type *nf1* alleles (*nf1a+/+;nf1b+/+;EGFP;MYCN*; *Figure 3*), indicating derivation from sympathetic neuroblast precursors (TH+, Hu+). This is consistent with our previous report in *nf1* wild-type fish that *MYCN*-induced neuroblastoma tumors arise from adrenal sympathetic neuroblasts that are prevented from differentiation into chromaffin cells by the overexpression of *MYCN* (*Zhu et al., 2012*). Predominately TH+, Hu- chromaffin cells were detected in the IRG of *nf1a* mutant fish lacking *MYCN* overexpression (*Figure 3—figure supplement 1*), indicating that loss of *nf1* alone does not contribute to developmental arrest of PSNS neuroblasts at the TH+, Hu+ stage. Taken together, these data demonstrate that loss of *nf1a* synergizes with overexpression of *MYCN* in the initiation of neuroblastoma, but that loss of *nf1a* is insufficient to initiate neuroblastoma on its own, possibly because loss of *nf1a* by itself does not lead to a block of terminal differentiation in PSNS neuroblasts in the IRG.

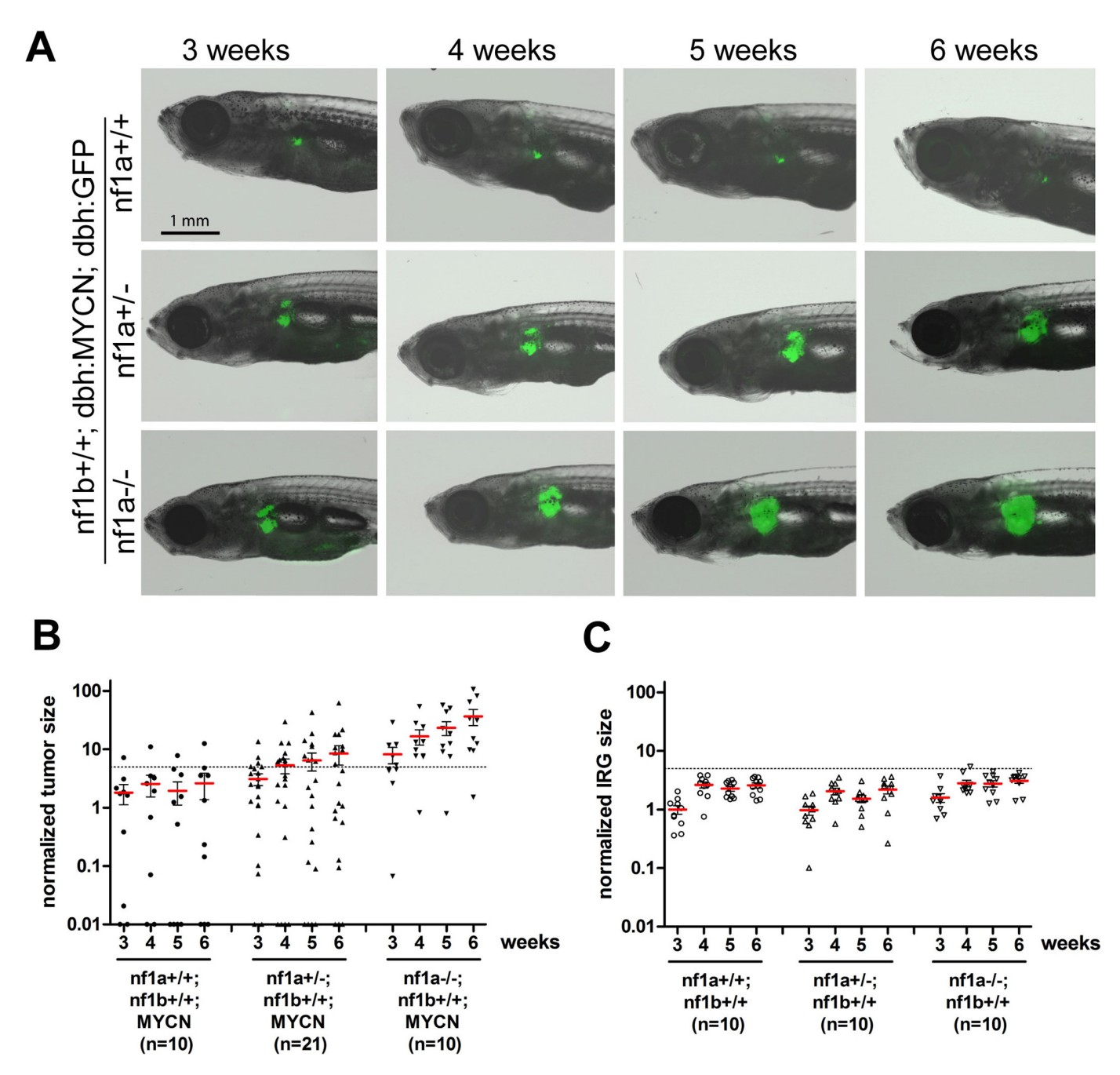

**Figure 4.** Loss of *nf1* promotes neuroblastoma tumor progression in *MYCN* transgenic fish. (**A**) Neuroblastoma development in representative fish of the *nf1a+/+;nf1b+/+;MYCN;GFP, nf1a+/-;nf1b+/+;MYCN;GFP* and *nf1a-/-;nf1b+/+;MYCN;GFP* genotypes over 3 to 6 weeks of age.Each fish was imaged weekly for 3 continuous weeks from 3 weeks of age. (**B**) Quantification of GFP+ sympathoadrenal cells in the IRG of fish with *nf1a+/+;nf1b+/+; MYCN;GFP* (n = 10), *nf1a+/-;nf1b+/+;MYCN;GFP* (n = 21) and *nf1a-/-;nf1b+/+;MYCN;GFP* (n = 15) genotypes, demonstrating tumor progression over 3 weeks. Tumors were scored when the GFP+ signals in the IRG exceeded the threshold defined by the dotted line. (**C**) IRG development in *nf1a* mutant zebrafish lacking overexpression of *MYCN*. Quantification of GFP+ sympathoadrenal cells in the IRG of fish with *nf1a+/+;nf1b+/+;GFP* (n = 10), *nf1a+/-; nf1b+/+;GFP* (n = 10) and *nf1a-/-;nf1b+/+;GFP* (n = 10) genotypes. The same tumor threshold line shown in panel B is included for comparison.

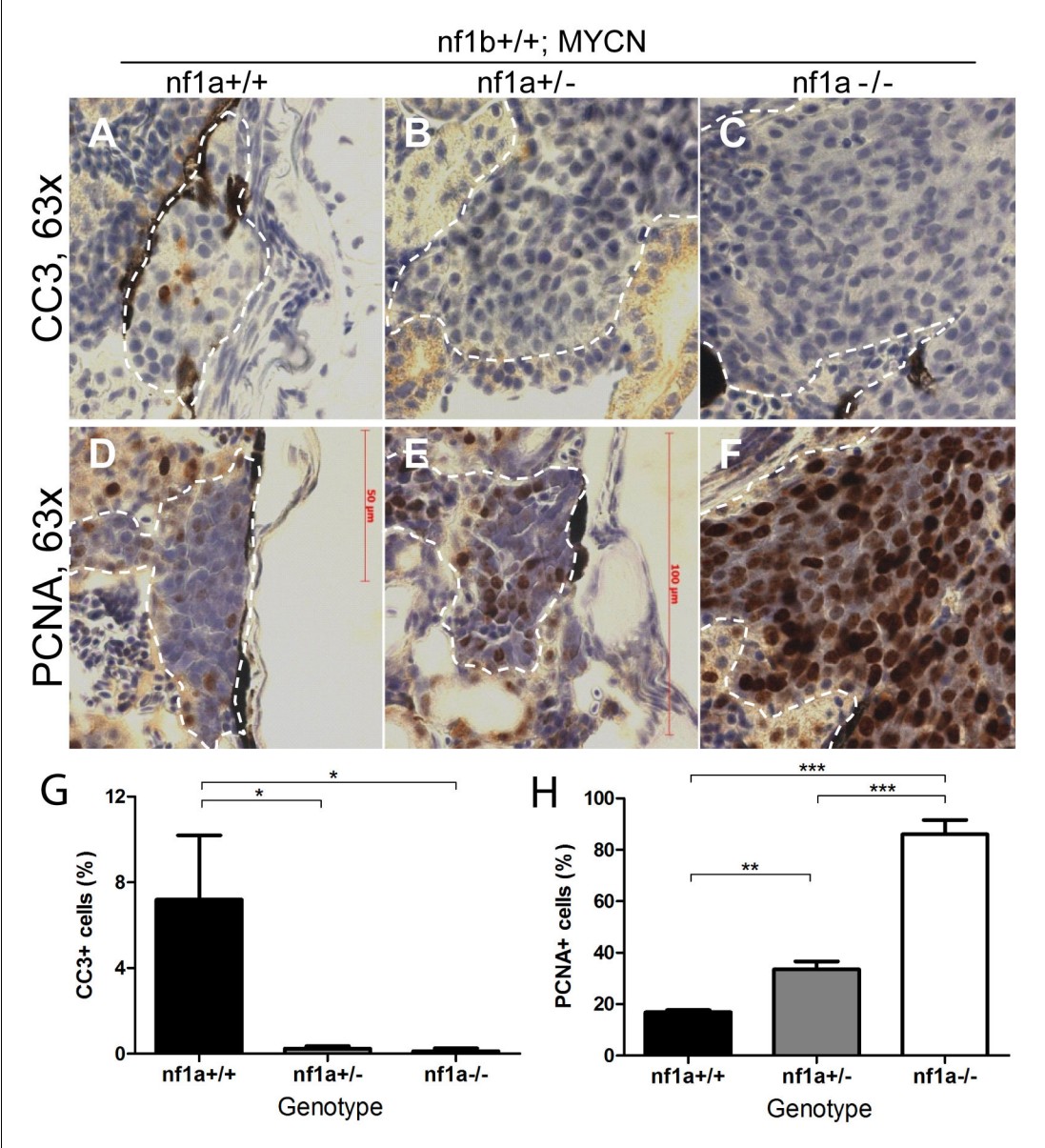

**Figure 5.** Loss of *nf1* suppresses apoptosis and enhances proliferation of tumor cells in MYCN-driven neuroblastoma. Immunohistochemical analysis of sagittal sections through tumors in the IRG of *nf1a+/+;nf1b+/+;MYCN;GFP* (A,D), *nf1a+/-;nf1b+/+;MYCN;GFP* (B,E) and *nf1a-/-;nf1b+/+;MYCN;GFP* (C, F) fish at the age of 6 weeks, using antibodies against cleaved caspase-3 (CC3; A–C) and proliferating cell nuclear antigen (PCNA; D-F).Dotted lines indicate tumor boundaries. Quantification of CC3-positive (G) and PCNA-positive (H) cells. Values are means + s.e.m. per section (three *nf1a+/+;nf1b+/+;MYCN;GFP,* five *nf1a+/-;nf1b+/+;MYCN;GFP* and four *nf1a-/-;nf1b+/+;MYCN;GFP* tumors). *p<0.05, **p<0.01, ***p<0.001 by two-tailed unpaired t-test.

The following figure supplement is available for figure 5:

**Figure supplement 1.** Quantitative RT-PCR showing the relative expression levels of key genes involved in control of the cell cycle in neuroblastoma tumors with wild-type or mutant *nf1a*.

## Loss of *nf1* promotes MYCN-induced neuroblastoma progression in vivo in a dose-dependent manner

Taking advantage of the optical transparency of our zebrafish neuroblastoma model, which allows us to monitor fluorescent tumor cell progression in vivo, we further investigated the compound effects

of loss of *nf1* and gain of *MYCN* on the kinetics of neuroblastoma progression. For these studies, we focused on loss of *nf1a*, because this orthologue is the most highly expressed (*Figure 1F*), and its loss synergizes most prominently with overexpression of *MYCN* during tumorigenesis (*Figure 2C*). Expansion of EGFP+ sympathoadrenal cells was observed in the IRG of *nf1a* heterozygous zebrafish (*nf1a+/-;nf1b+/+;MYCN;EGFP*, *Figure 4A* middle panel), with an average increased expansion of 1.73 fold by the end of the 3-week period and a tumor induction rate of 43% (9/21 fish had tumors before they reached 6 weeks of age; *Figure 4B*). By contrast, the EGFP+ sympathoadrenal cells in the IRG of *nf1a-/-* homozygous zebrafish expanded much more rapidly, with an average 6.61-fold expansion rate within the 3 week period (*nf1a-/-;nf1b+/+; MYCN;EGFP*) and tumor development in 9 of 10 fish before 6 weeks of age (*Figure 4B*). These data demonstrate that the loss of *nf1a* promotes MYCN-induced neuroblastoma onset and progression in vivo in zebrafish in a dose-dependent manner.

## Loss of *nf1a* synergizes with MYCN overexpression by promoting both tumor cell survival and proliferation

Wild-type zebrafish transgenic for *MYCN* exhibit hyperplasia of sympathetic neuroblast precursors in the IRG from 3 to 5 weeks of life, followed by a developmentally-timed apoptotic response, resulting in regression of the hyperplastic sympathoadrenal cells in most fish. This leads to a low penetrance, with tumors developing in 515% of animals, and a relatively long latency of 12 to 20 weeks (*Figure 2C*) (*Zhu et al., 2012*). In the current study, we observed positive staining for cleaved caspase-3 at 6 weeks of age in *nf1* wild-type *MYCN* transgenic fish, consistent with our previous findings, but we did not detect staining for cleaved caspase-3 in the IRG of either *nf1a+/-;nf1b+/+; MYCN;EGFP* or *nf1a-/-;nf1b+/+;MYCN;EGFP* fish (*Figure 5*). Thus, loss of either one or both alleles of *nf1a* blocked the apoptotic response due to overexpression of MYCN in sympathetic neuroblasts, thus promoting tumor cell survival and leading to the development of neuroblastoma at high penetrance (*Figure 2*). The effects of loss of NF1 on cell survival therefore appear similar to the effects of mutational activation of the ALK tyrosine kinase (*Zhu et al., 2012*), which also promotes activated signaling through the RAS pathway.

We also analyzed the effect of *nf1* loss on the proliferative capacity of *MYCN*-overexpressing sympathetic neuroblast precursors by evaluating the fraction of cells expressing the proliferating cell nuclear antigen (PCNA). We found that neuroblastomas arising in the *nf1a+/-* and *nf1a-/-* background exhibited a progressively higher percentage of nuclei with PCNA staining (30% for *nf1a+/-* and 80% for *nf1a-/-*) compared to PCNA staining of 20% of nuclei in the wild-type *nf1a+/+* genotype (*Figure 5*). Thus, loss of *nf1a* resulted in an increased neuroblastoma cell proliferative fraction, consistent with the observation that neuroblastoma tumors grow much more rapidly in fish with homozygous loss of *nf1a* (*Figure 4B*). Together, our results support an important role of the tumor suppressor *nf1* in restricting both the proliferative rate and survival of *MYCN*-overexpressing hyperplastic sympathetic neuroblast precursors. They also provide important cellular mechanisms that underlie the striking synergy between *nf1* loss and overexpression of *MYCN* in neuroblastoma pathogenesis. We then isolated established neuroblastoma tumors of *nf1a+/+;nf1b+/+;MYCN* and *nf1a-/-;nf1b+/+;MYCN* zebrafish to examine expression of key genes involved in control of the cell cycle, including *ccna1, ccna2, ccnb1, ccnd1, ccnd2, ccnd3, ccne, cdk2, cdk4, cdk6* and *e2f1*. We did not detect significant differences in mRNA levels of these genes (*Figure 5—figure supplement 1*), suggesting that further experiments are required to decipher the molecular mechanism through which the loss of nf1a promotes increased proliferation of MYCN-overexpressing neuroblastoma tumor cells.

## Loss of *nf1* results in aberrant activation of RAS signaling in *MYCN*-induced neuroblastoma in vivo

Given the well-described role of neurofibromin as a negative regulator of RAS signaling, we postulated that *nf1a* loss in our neuroblastoma model would lead to activation of effector pathways downstream of RAS. To test this hypothesis, we first assessed the MYCN-independent activation of RAS effector pathways in the IRG of *nf1*-deficient fish by immunohistochemistry (wild-type *nf1a+/+;nf1b +/+;EGFP* fish *vs. nf1a+/-;nf1b+/+;EGFP* and *nf1a-/-;nf1b+/+;EGFP* fish). The IRG of wild-type *nf1a +/+;nf1b+/+;EGFP* fish did not exhibit detectable phosphorylated ERK (pERK), phosphorylated AKT

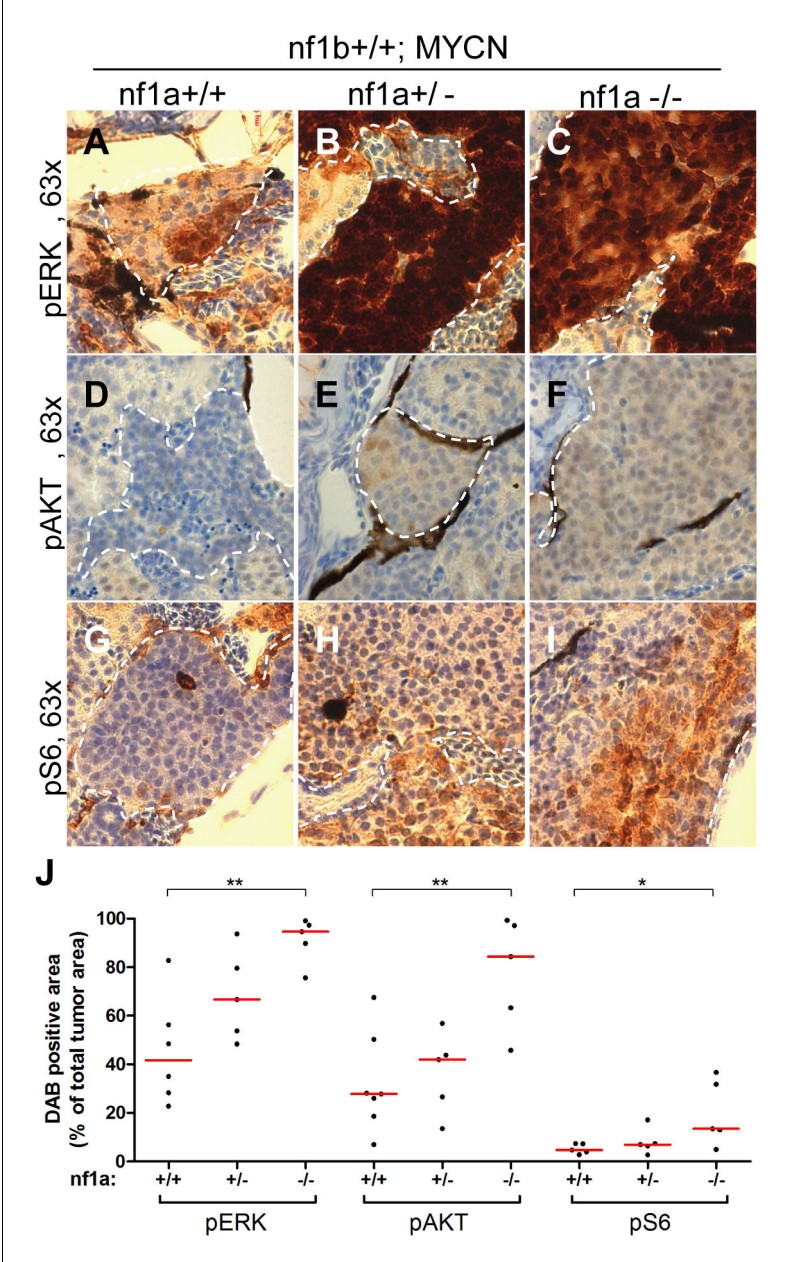

**Figure 6.** Loss of *nf1* results in aberrant ERK/Akt/mTOR signaling in MYCN-driven neuroblastoma. Immunohistochemical analysis of sagittal sections through tumors in the IRG of *nf1a+/+;nf1b+/+;MYCN;GFP* (A,D, G), *nf1a+/-;nf1b+/+;MYCN;GFP* (B,E,H) and *nf1a-/-;nf1b+/+;MYCN;GFP* (C,F,I) fish at the age of 6 weeks, using antibodies against phosphorylated ERK1/2 (pERK, A–C), phosphorylated AKT (pAKT, D–E) and phosphorylated S6 (pS6, G–I). Dotted lines indicate the tumor boundaries. The quantification of pERK-, pAKT- and pS6-positive tumor areas are shown in (J), with the red bars representing the median values. ns p>0.05, *p<0.05, **p<0.01 by two-tailed unpaired t-test.

The following figure supplement is available for figure 6:

**Figure supplement 1.** No aberrant ERK/AKT/mTOR signaling was detected in the IRG of nf1a mutant fish that lacked MYCN overexpression.

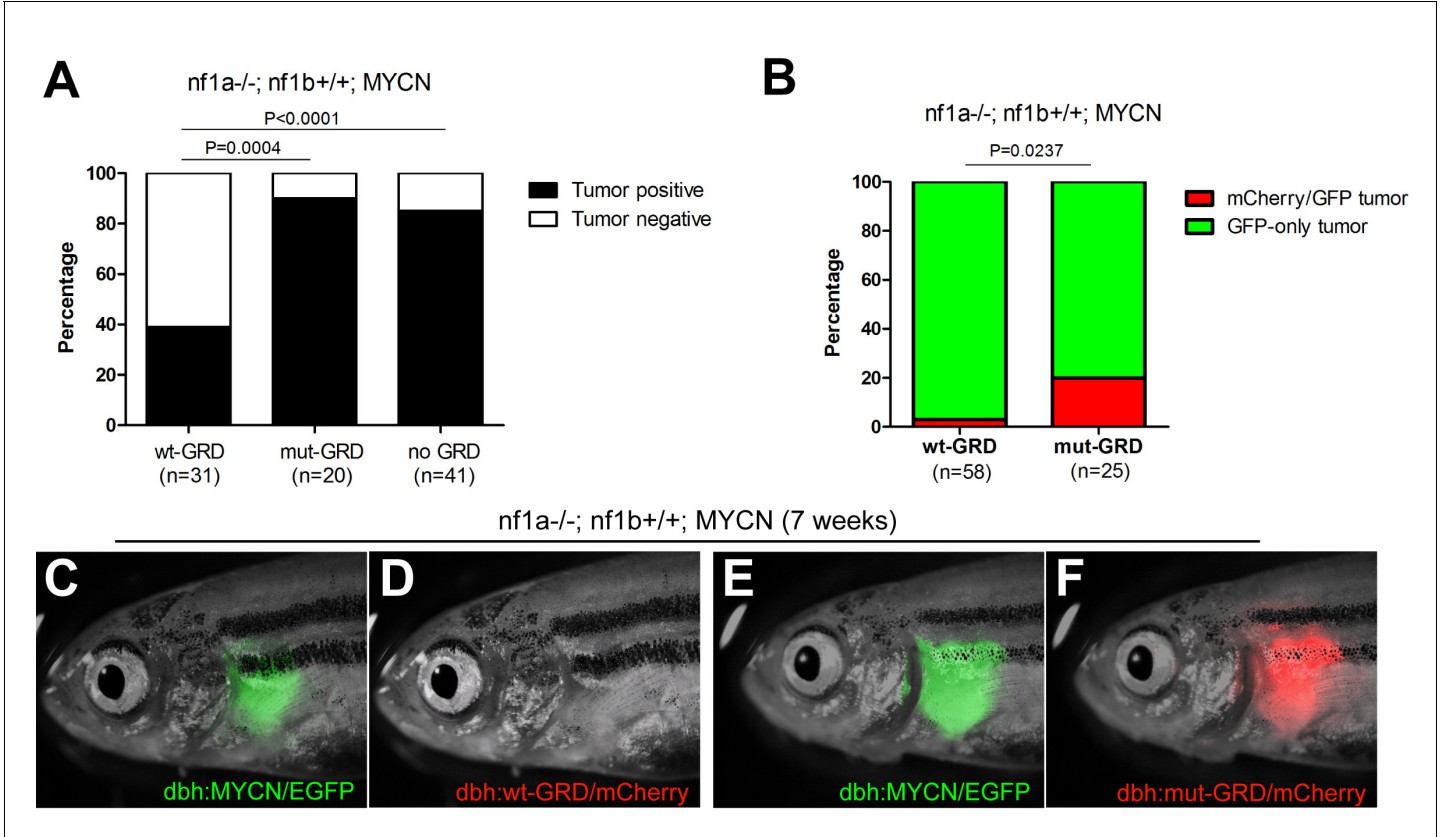

**Figure 7.** GRD domain is required for the tumor suppressor function of *nf1* in *MYCN*-driven neuroblastoma tumorigenesis. (**A**) Tumor penetrance of stable transgenic zebrafish with the genotypes of *nf1a-/-;nf1b+/+;GFP;wt-GRD;mCherry* (n = 31), *nf1a-/-;nf1b+/+;GFP;mut-GRD;mCherry* (n = 20) and *nf1a-/-;nf1b+/+;GFP;mCherry* (n = 41) at the age of 12 weeks. (**B**) *nf1a-/-;nf1b+/+;MYCN;GFP* fish injected with *dbh:wt-GRD;dbh:mCherry* (n = 58) showed a significantly lower *mCherry+* mosaic tumor rate compared with the sibling fish injected with *dbh:mut-GRD;dbh:mCherry* (n = 25) at the age of 7 weeks (Fisher's exact test), indicating the tumor suppression function of the NF1 GRD domain.Most early tumors arose in *nf1a-/-;nf1b+/+;MYCN;GFP* fish injected with *dbh:wt-GRD; dbh:mCherry* only expressed GFP (**C**) but not mCherry fluorescent protein (**D**). A significant subset of early tumors arose in *nf1a-/-;nf1b+/+;MYCN;GFP* fish injected with *dbh:mut-GRD; dbh:mCherry* did express both GFP (**E**) and mCherry (**F**).

The following figure supplement is available for figure 7:

**Figure supplement 1.** Scheme of the mosaic structure-function analysis using NF1-GRD.

(pAKT) or phosphorylated S6 (pS6), demonstrating a very low level of basal RAS activity in the normal sympathoadrenal cells of the IRG (*Figure 6—figure supplement 1*). We did not observe notable increases in pERK, pAKT or pS6 in the nontransformed IRG of *nf1a* mutant fish that lacked *MYCN* overexpression (*Figure 6—figure supplement 1*). Thus, the loss of *nf1* did not cause a detectable increase in the basal levels of endogenous RAS signaling in normal sympathoadrenal cells of the IRG, probably because in the absence of MYCN overexpression they rapidly differentiate into mature chromaffin cells (*Zhu et al., 2012*).

We then assessed the role of *nf1* in suppressing RAS effector pathways in the neuroblastoma cells of wild-type fish overexpressing *MYCN*. By immunohistochemistry and measurement of the area of the diaminobenzidine-peroxidase (DAB)-stained tumor cells, we detected high levels of pERK, pAKT and pS6 in neuroblastomas from 6-week-old fish (*Figure 6A,D,G and J*), indicating activation of RAS effector pathways in the proliferating neuroblasts of wild-type *nf1a+/+; nf1b+/+;MYCN* fish that were blocked from terminal differentiation by overexpression of MYCN. In this context, it is not surprising that we observed much higher levels of activation of these effector pathways in neuroblastomas arising in *nf1a*-deficient fish, determined by significant increases of positive DAB-stained tumor areas (*Figure 6 B,C,E,F,H,I and J*). Thus, the overexpression of MYCN blocks the differentiation of sympathoadrenal precursor cells, leaving them vulnerable to the potentiating effects of *nf1* loss on

RAS pathway activation, which in turn leads to the marked increased in neuroblastoma penetrance and growth rate.

## The GAP activity of GRD domain is sufficient for tumor suppression by *NF1* in neuroblastoma

To test the ability of the GRD domain to rescue tumor suppression in neuroblastoma, we bred the *nf1a-/-;nf1b+/+;dbh:wt-GRD;dbh:mCherry* transgenic line (as in *Figure 1H*) with the *nf1a-/-;nf1b+/+; MYCN;EGFP* fish. Only 12 of 31 *nf1a-/-;nf1b+/+;MYCN;EGFP;wt-GRD;mCherry* fish developed neuroblastoma tumors by the age of 12 weeks (38.7%, *Figure 7A*), compared to 35 of 41 *nf1a-/-;nf1b+/+;MYCN;EGFP* fish (85.4%, p<0.0001; *Figure 7A*), indicating that restoration of the NF1 GAP activity significantly reduced the penetrance of neuroblastoma in the *nf1a-/-;nf1b+/+;MYCN;EGFP* background. As a control, we also stably expressed the inactive GRD R1276P mutant (designated 'mut-GRD') into *nf1a-/-;nf1b+/+;MYCN;EGFP* fish, which led to a frequency of neuroblastoma at 12 weeks of age that was identical to that in fish not transgenic for either GRD construct (90.0%, *Figure 7A*). Because the GRD R1276P mutant specifically disrupts the ability of the GRD to mediate the NF1 GAP activity, this result indicates that NF1 normally suppresses the development of neuroblastoma by downmodulating oncogenic signals from RAS. This finding stands in marked contrast to the inability of the same zebrafish line expressing the same level of GRD to rescue overgrowth of the SCG in normal development, indicating that even though a novel mechanism is responsible for restriction of PSNS neuron growth during normal embryogenesis, NF1 still acts as a classical GAP protein to suppress RAS-MAPK signaling for the suppression of neuroblastoma.

To control for possible founder effects in our stable transgenic lines, we constructed primary mosaic GRD transgenic fish by coinjecting the *dbh:wt-GRD* or *dbh:mut-GRD* with the *dbh:mCherry* fluorescent marker into one-cell stage *nf1a-/-;nf1b+/+;MYCN;EGFP* embryos. Coinjected constructs integrate together in cells of the primary injectant, producing fish with mosaic coexpression of the transgenes (*Langenau et al., 2008*). The MYCN-driven neuroblastomas that arise in these fish express the EGFP that is fused to MYCN, but mCherry is detected only if the tumors arise from a precursor cell that has integrated and expresses the *mCherry* and *GRD* transgenes (*Figure 7figure supplement 1*). As shown in *Figure 7B, D and F*, only 2 of 58 neuroblastoma tumors in fish coinjected with the functionally active wt-GRD domain and the *mCherry* transgene exhibited mCherry fluorescence at 7 weeks of age (3.5%, *Figure 7B*). By contrast, coinjection of the mut-GRD domain and the *mCherry* transgene into *nf1a-/-;nf1b+/+;MYCN;EGFP* fish yielded strong mCherry fluorescence in 5 of 25 tumors that developed by 7 weeks of age (20.0%; p=0.0237; *Figure 7B, D, and F*). This experiment in primary injectants confirms our results in stable GRD transgenic lines, demonstrating that NF1 functions differently in normal development and tumorigenesis of the PSNS, and the tumor suppression function of NF1 in MYCN-induced neuroblastoma is mediated through the GAP activity of the GRD domain.

## Inhibition of RAS-MAPK signaling sensitizes *nf1*-deficient neuroblastoma to retinoic acid in vivo

Because very aggressive growth properties of MYCN-induced neuroblastomas in zebrafish with loss of both alleles of *nf1a* (See *Figure 2*) are attributable to aberrant activation of RAS-MAPK signaling (see *Figures 6* and *7*), we evaluated the importance of MEK inhibition for the treatment of these tumors. For these experiments, we used the FDA-approved MEK inhibitor trametinib to treat primary neuroblastoma tumors that were growing in 3-week-old *nf1a-/-;nf1b+/+;MYCN; EGFP* fish (*Figure 8*). After 1 week of treatment with trametinib, we observed that the drug significantly reduced the rate of tumor growth, but did not by itself cause shrinkage of the tumor (*Figure 8A*).

The real impact of inhibition of RAS-MAPK signaling by trametinib was evident when it was tested together with isotretinoin (*Figure 8*). Isotretinoin, also known as 13-cis retinoic acid, is a retinoid used as frontline therapy for childhood neuroblastoma and improves the event-free survival of combination chemotherapy (*Matthay et al., 2009*; *Park et al., 2009*). In our *nf1a-/-;nf1b+/+;MYCN; EGFP* fish, isotretinoin alone leads to reduced tumor growth, with essentially stable tumor size over 7 days of treatment at the highest dosages (*Figure 8A*). However, the combination of isotretinoin and trametinib was much more active in reducing tumor size, in many fish reducing the GFP fluorescence from the *dβh* promoter to the level of the normal IRG at 4 weeks of life (*Figure 8A and C*).

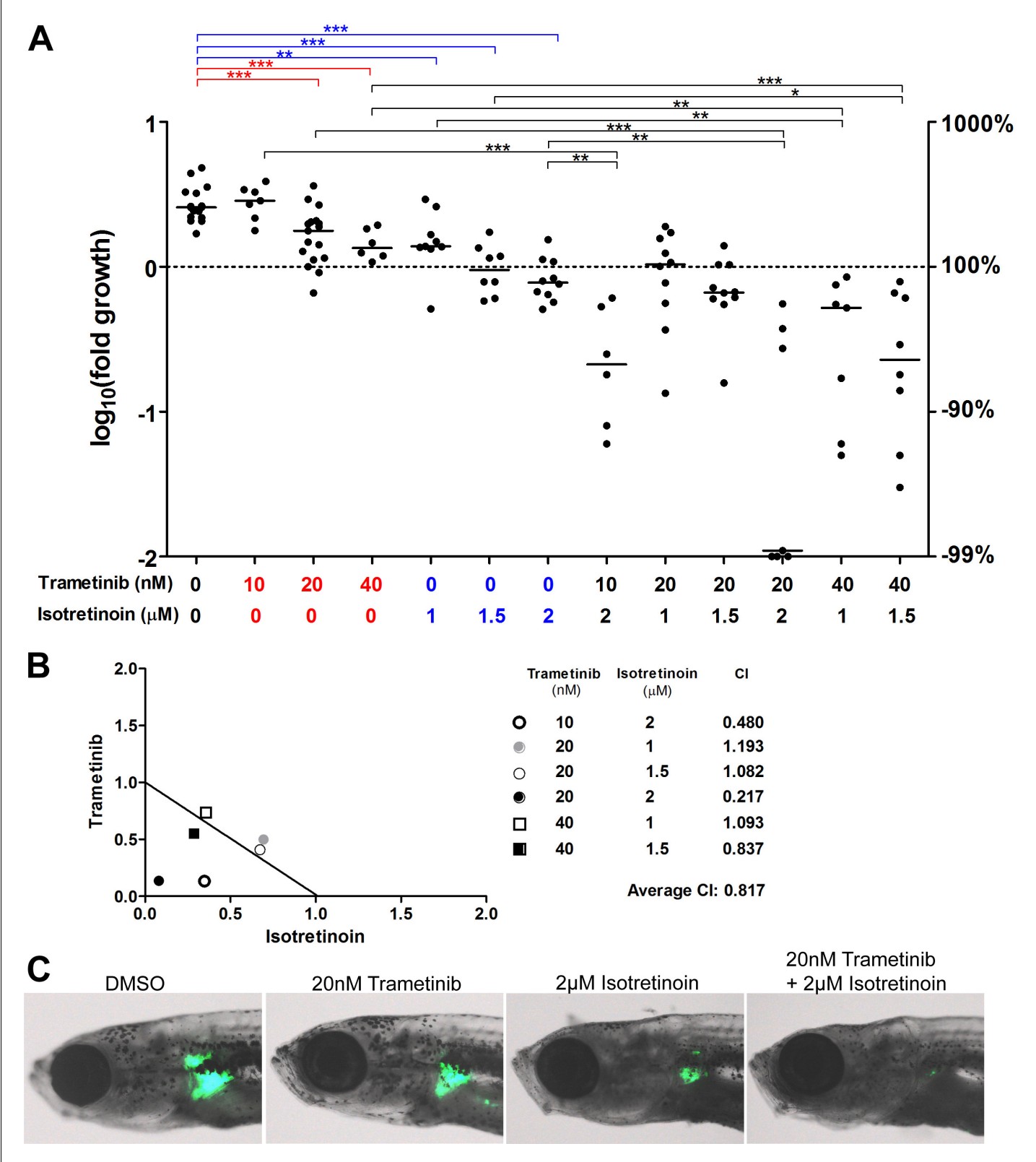

**Figure 8.** The MEK inhibitor trametinib synergizes with isotretinoin in suppressing nf1-deficient neuroblastoma in vivo. (**A**) Neuroblastoma growth in *nf1a-/-;nf1b+/+;MYCN;GFP* fish treated with vehicle control, trametinib, isotretnoin or combinations of trametinib and isotretnoin. Representative fish are shown in (**C**). (**B**) Synergistic effects of trametinib and isotretinoin on tumor suppression were analyzed by isobologram analysis.

Importantly, synergistic antitumor effects of trametinib and isotretinoin were documented in vivo at several different dosage combinations by isobologram analysis (*Figure 8B*), indicating that inhibition of RAS-MAPK signaling can significantly improve the treatment of this very aggressive form of neuroblastoma when it is combined with the inhibition of other key pathways. Because of the very high penetrance and rapid onset of neuroblastoma in our *nf1*-deficient, MYCN-transgenic zebrafish model, it becomes one of a very few systems in which extensive analysis of the synergistic activity of two or more drugs can be evaluated in primary tumors in vivo. This capability is especially valuable given the recent evidence documenting the profound impact of RAS-MAPK hyperactivation in therapy-resistant neuroblastoma of childhood (*Eleveld et al., 2015*).

## Discussion

In this study, we use a zebrafish model to demonstrate that NF1 loss can potentiate the tumorigenic effects of *MYCN*-overexpression in high-risk neuroblastoma. We show that *nf1* deficiency in the fish leads to aberrant activation of RAS-MAPK signaling in MYCN-induced neuroblastoma, which promotes both tumor cell survival and proliferation, leading to marked acceleration of tumor onset and increased tumor penetrance. To establish the underlying mechanism of these observations, we performed in vivo structure-function analyses to show that the tumor suppression function of NF1 is mediated through the GAP activity of its GRD domain. The same rescue construct does not suppress the PSNS overgrowth that accompanies *nf1* loss during development, supporting earlier work implicating a different and still-to-be-identified activity of NF1 in the restriction of the normal growth of PSNS neuroblasts during embryogenesis.

We and others have shown that inactivation of the *NF1* gene in human neuroblastoma and melanoma cell lines does not result in increased levels of the GTP-bound form of Ras compared to similar lines that express normal levels of the NF1 protein (*Johnson et al., 1993*; *The et al., 1993*). Ostensibly, this would seem inconsistent with our current findings in the zebrafish model implicating down-regulation of the RAS pathway in NF1-mediated tumor suppression. However, a recent whole-genome sequencing study revealed that loss of *NF1* is one of a strikingly high frequency of diverse mutations that hyperactivate the RAS-MAPK signaling pathway in relapsed neuroblastoma (*Eleveld et al., 2015*). Thus, our inability to demonstrate increased GTP on RAS in *NF1*-deficient neuroblastoma cell lines may reflect the strong activation of the RAS-MAPK signaling pathway in essentially all high-risk neuroblastomas, if not by *NF1* loss then by direct mutation of *RAS* itself or other types of mutations. This interpretation becomes more appealing when one realizes that neuroblastomas able to grow in vitro as continuous cell lines are invariably from patients with the very highest risk of treatment failure, and often are obtained from tumor biopsies at the time of relapse. The same relationship likely applies to melanoma cell lines as well, since most of these tumors harbor activating mutations in the RAS-MAPK pathway, including *BRAF* V600E and *NRAS* Q61L/R, as well as the loss of *NF1* in a subset of cases (*Cancer Genome Atlas Network, 2015*; *Hodis et al., 2012*).

Studies of neural crest and PSNS development in *Nf1* null mice were the first to implicate a GAP-independent activity of *Nf1* in restricting the growth of sympathoadrenal cells, as the GRD domain was not able to rescue the overgrowth of sympathetic neuroblasts in the adrenal medulla (*Ismat et al., 2006*). Our results support those studies, in that the GRD domain expressed in the same zebrafish line that suppressed neuroblastoma pathogenesis failed to suppress sympathetic neuroblast overgrowth in *nf1*-deficient zebrafish embryos. Thus, GAP-independent activities of *NF1* appear to restrict the growth of normal neural crest-derived tissues during development, as the PSNS overgrowth phenotype was not rescued by the same level of GRD restoration that suppressed the accelerated onset of MYCN-driven neuroblastoma (*Figures 1* and *7*).

The recent discovery that mutations leading to activated RAS-MAPK signaling occur in nearly 80% of relapsed neuroblastomas has clear implications for the selection of targeted therapy for this tumor (*Eleveld et al., 2015*). Indeed, it suggests that MEK inhibitors and other agents targeting the RAS-MAPK pathway might suppress the outgrowth of resistant clones and therefore should be investigated as part of the initial combination therapy for high-risk neuroblastoma. Our zebrafish model of MYCN-driven neuroblastoma with *nf1* loss is ideally suited for the rapid in vivo analysis of the effects of candidate small-molecule inhibitors selected for this purpose. Neuroblastomas can be detected in most MYCN-transgenic fish with *nf1a* loss by 3 weeks of age, when the fish are very small, making it feasible to test the effectiveness of many drugs and drug combinations for their

ability to kill primary neuroblastoma cells in vivo. This advantage of the model is readily apparent in our demonstration of marked antitumor synergy between the MEK inhibitor trametinib and the retinoid isoretinoin in *MYCN*-overexpressing fish with loss of nf1 (*Figure 8*). Studies using our model should be a valuable asset in devising new therapeutic strategies for neuroblastomas with mutations affecting the RAS-MAPK pathway, which appear to be a major cause of relapse in children with this devastating tumor.

## Material and methods

### Zebrafish

The previously described *nf1a* and *nf1b* mutant zebrafish lines (*Shin et al., 2012*) were crossed with transgenic lines including Tg(*dbh:EGFP*) and Tg(*dbh:EGFP-MYCN*) (*Zhu et al., 2012*) for this study. The compound mutant fish were fin-clipped and genotyped for the *nf1a* and *nf1b* as previously described (*Shin et al., 2012*). All zebrafish studies and maintenance of the animals were performed in accordance with Dana-Farber Cancer Institute IACUC-approved protocol #02–107.

### DNA constructs for transgenesis

The DNA construct for *dbh:wt-GRD, dbh:mut-GRD* and *dbh:mCherry* was subcloned using the Multisite Gateway System (Invitrogen, Carlsbad, CA) as previously described (Invitrogen) (*Ismat et al., 2006*; *Zhu et al., 2012*). Embryos were injected with these DNA constructs at the one-cell stage as previously described (*Zhu et al., 2012*) and grown to adulthood. Fin clips from the offspring were genotyped for the stable integration and germline transmission of the transgenes.

### Neuroblastoma tumor watch

*nf1a+/-; nf1b+/-; dbh:EGFP* and *nf1a+/-; nf1b+/-; dbh:EGFP-MYCN* mutant zebrafish were crossed, and offspring were screened every 2 weeks, starting from the age of 4 weeks, for fluorescent EGFP expressing cell masses indicative of tumors (*Zhu et al., 2012*). Once an EGFP-positive cell mass was identified, the individual fish were separated and carefully monitored weekly for at least 4 weeks for tumor progression. Only the fish with progressing EGFP-positive cell masses were scored as tumor fish and analyzed further by H&E staining and immunohistochemical assays. All fish were genotyped for *nf1a* and *nf1b* at the age of 8 weeks.

### Cell sorting, RNA extraction and quantitative RT-PCR

Wild-type *dbh:mCherry* embryos (n = 2000) were collected at the age of 1 week, and dissociated into single-cell suspensions using 0.05% trypsin-EDTA (Life Technologies, Carlsbad, CA). Cells were filtered through a 40-μm cell strainer (Falcon, Corning, NY) and resuspended in PBS. Fluorescence-activated cell sorting was performed on a BD FACSAria II CORP UV (Dana-Farber Cancer Institute Hematologic Neoplasia Flow Cytometry Core), and *mCherry*-positive cells, as well as the *mCherry*-negative control cells, were collected for RNA extraction. For juvenlie zebrafish at the ages of 3, 4, and 6 weeks, 5 wild-type fish at each age were homogenized in a tissue grinder for RNA extraction. Total RNA was extracted with Trizol reagent (Invitrogen) and purified with the Qiagen RNeasy kit (Qiagen, Santa Clarita, CA) according to the manufacturer's instructions. cDNA was synthesized from total RNA using the iScript cDNA synthesis kit (Biorad, Hercules, CA) according to the manufacturer's instructions (Biorad). The Q-RT-PCR reactions were performed with the SYBR green PCR Core Reagents kit (Applied Biosystems, Foster City, CA) and ViiA 7 Real-Time PCR System (Life Technologies) according to the manufacturer's instructions. All Q-RT-PCR assays were performed in triplicate, with β-actin used as an endogenous control. Q-RT-PCR primer sequences spanning exon-exon junctions were as follows: *nf1a* forward, 5'-AAATTCCAGACTACGCCGAGC-3' and reverse, 5'-TATAAACTATAGGGCCCTCTGGGGA-3'; *nf1b* forward, 5'- TGGCGCAGAAGTTTGCATTTCAATA -3' and reverse, 5'- GCAATGACTGTGGCTTCGATT-3'; *β-actin* forward, 5'- TTCCTGGGTATGGAATC TTGCG -3' and reverse, 5'- GTGGAAGGAGCAAGAGAGGTG -3'; *ccna1* forward, 5'- TGGC TCAGGGTCATTTATGG-3' and reverse, 5'- TAACTTCGCATTCACGCAGG-3'; *ccna2* forward, 5'-TCCACTGGAGGCCAGTTTTG-3' and reverse, 5'- GACTTGACCTCCATTTCCCG-3'; *ccnb1* forward, 5'-ATTCTCCTCAGTGTTTCTCCAGTC-3' and reverse, 5'-AAGTGTAGATGTCTCGCTCATATTC-3'; *ccnd1* forward, 5'-TTGCTGCGAAGTGGATACCATAAG-3' and reverse, 5'-AGGCACAATTTCTTTC

TGAACACAC-3'; *ccnd2* forward, 5'- CCGTCCTGATCCGAATCTTCTG-3' and reverse, 5'-GCCACCA TCCTCCGCATAAAG-3'; *ccnd3* forward, 5'-ACGGCTACAGAGCTGAAGTT-3' and reverse, 5'-CATC TGCTCGGCGCTAACA-3'; *ccne* forward, 5'-ACAACCTGCTCGGAAAAGACAAG-3' and reverse, 5'- CACAAACCTCCATTAGCCAGTCC-3'; *cdk2* forward, 5'-TCGCGCTGAAGAAAATCCGA-3' and reverse, 5'-ACGCAACTTGACTATGTTAGGGT-3'; *cdk4* forward, 5'-TGAGCCAGTAGCAGAGATCG-3' and reverse, 5'-AGTGGGAGTCCGTCCTGATT-3'; *cdk6* forward, 5'-TCTCACCGTGTGGTTCATCG-3' and reverse, 5'-ATGTCACAACCACCACGGAA-3'; *e2f1* forward, 5'-ACAACATCCAGTGGC TAGGG-3' and reverse, 5'-TTCGTCCAGTTTCTCCTCGG-3'.

## Immunohistochemistry

Zebrafish were euthanized in tricaine anesthetic, fixed in 4% paraformaldehyde at 4°C for 2 days, and decalcified with 0.25 M EDTA, pH 8.0, for at least 24 hr. Paraffin sectioning followed by hema- toxylin and eosin (H&E) staining or immunohistochemistry (IHC) was performed at the Dana-Farber/ Harvard Cancer Center Research Pathology Core. Primary antibodies included Phospho-p44/42 MAPK (ERK1/2) (Thr202/Tyr204, Cell Signaling #4370; 1:150), Phospho-AKT (Ser473, Cell Signaling #4060), Phospho-S6 ribosomal protein (Ser240/244, Cell Signaling #4838), PCNA (PC10, EMD Milli- pore; 1:100), cleaved Caspase-3 (Cell Signaling #9664; 1:250), TH (Pel-Freez # P40101, 1:500) and HuC/D (Invitrogen #A-21271, 1:200). Antibody binding was detected with a diaminobenzidine-per- oxidase (DAB) visualization system (EnVision+, Dako, Carpinteria, CA). Mayer's hematoxylin was used for counterstaining.

## Imaging and quantification

For brightfield DIC images, a Zeiss Axio Imager.Z1 compound microscope equipped with an Axio- Cam HRc was used. For quantification of immunohistochemistry staining, brightfield images were taken with the Mantra Quantitative Pathology Workstation (Perkin Elmer, Norwalk, CT) and analyzed with ImageJ. The color deconvolution plugin of ImageJ and the 'H DAB' vector was used to separate the hematoxylin and DAB stains (*Ruifrok and Johnston, 2001*), and the kidney tubules adjacent to the IRG or tumor region were applied as internal references to define threshold of DAB staining for each IRG or tumor region. For live imaging, zebrafish and embryos were anaesthetized using 0.016% tricaine () and mounted in 4% methycellulose (). A Nikon SMZ1500 microscope equipped with a Nikon digital sight DS-U1 camera was used for capturing both the bright field and fluorescent images from live zebrafish and embryos. For PSNS and neuroblastoma quantification, all animals in the same experiments were imaged under the same conditions and the acquired fluorescent images were quantified using the ImageJ software by measuring the EGFP covered area. For neuroblastoma quantification, the fluorescent area was normalized against the surface area of the fish head as fish size was variable. Overlays were created using ImageJ and Adobe Photoshop 7.0.1.

## Drug treatment

*nf1a-/-; nf1b+/+; dbh:EGFP; dbh:EGFP-MYCN* zebrafish with GFP+ tumor were imaged individually at the age of 3 weeks, separated and treated with trametinib () and/or isotretinoin (Selleck Chemi- cals, Houston, TX) with refreshment every 2 days. Drug synergism was evaluated using the CalcuSyn Software.

## Statistical analysis

Statistical analysis was performed with Prism 5 software (GraphPad). Kaplan-Meier methods and the log-rank test were applied to assess the rate of tumor development in *Figure 2*. Fish that died before they had evidence of EGFP-positive masses were censored. Fisher's exact test was used to assess the difference between tumor rate in wild-type versus mutant fish in *Figure 7*. A two-tailed unpaired t-test with confidence intervals of 95% was used for the analyses in *Figures 5*, *6* and *7*. The quantitative data in *Figures 4* and *7* are reported as means with standard errors of means (s.e.m). For *Figure 8*, a Mann Whitney test with confidence intervals of 95% was used for the analysis and the quantitative data are reported as median.

## Acknowledgements

We would like to thank John R Gilbert for critical review of the manuscript and editorial suggestions; Christian Lawrence for advice in zebrafish maintenance; Dr Donna S Neuberg for suggestions in statistical analysis, Dana-Farber/Harvard Cancer Center Research Pathology Core and Christine L Unitt for technical support.

## Additional information

### Funding

| Funder | Grant reference number | Author |
|---|---|---|
| U.S. Department of Defense | W81XWH-12-1-0125 | Jonathan A Epstein<br>A Thomas Look |
| Alex's Lemonade Stand Foundation for Childhood Cancer | | Shuning He<br>A Thomas Look |
| Children's Tumor Foundation | YIA | Shuning He |
| Rally Foundation | | Shuning He |
| Kay Kendall Leukaemia Fund | | Marc R Mansour |
| Leukaemia and Lymphoma Research | | Marc R Mansour |
| Damon Runyon Cancer Research Foundation | DRSG-9-14 | Mark W Zimmerman |
| National Institutes of Health | R01 CA180692 | A Thomas Look |

The funders had no role in study design, data collection and interpretation, or the decision to submit the work for publication.

### Author contributions

SH, Conception and design, Acquisition of data, Analysis and interpretation of data, Drafting or revising the article; MRM, ATL, Conception and design, Analysis and interpretation of data, Drafting or revising the article; MWZ, DHK, HML, KA, EG, Acquisition of data, Analysis and interpretation of data; EDdG, Acquisition of data, Contributed unpublished essential data or reagents; ARP-A, Analysis and interpretation of data, Drafting or revising the article; SZ, Analysis and interpretation of data, Contributed unpublished essential data or reagents; JAE, Conception and design, Contributed unpublished essential data or reagents

### Author ORCIDs

A Thomas Look, http://orcid.org/0000-0001-7851-8617

### Ethics

Animal experimentation: All zebrafish studies and maintenance of the animals were performed in accordance with Dana-Farber Cancer Institute IACUC-approved protocol (#02-107).

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
