## [Decision Letter]

Thank you for submitting your work entitled "Synergy Between Loss of NF1 and Overexpression of MYCN in Neuroblastoma Is Mediated by the GAP-related Domain" for consideration by *eLife*. Your article has been favorably evaluated by Sean Morrison (Senior editor) and two reviewers, one of whom, Richard Gilbertson, is a member of our Board of Reviewing Editors.

The reviewers have discussed the reviews with one another and the Reviewing Editor has drafted this decision to help you prepare a revised submission.

Summary:

Both reviewers stated that this is an important work that is well conducted, the conclusions of which are supported by the data presented.

Essential revisions:

Reviewer #2 presents four very interesting and pertinent points that could significantly improve the quality of the manuscript. Having reviewed these carefully, we suggest that the authors respond to the first and last of these four points, specifically:

1) How do levels of *nf1b* compare to *nf1a* after the first week of zebrafish development? Do the authors have any insights into how loss of *nf1a* drives proliferation here?

4) In Figure 6 and Figure 6—figure supplement 1, can data be quantified, especially given the result that *nf1a* loss does not seemingly affect RAS downstream signaling in non-MYCN amplified cells = no effect on endogenous levels in normal sympathoadrenal cells which the authors speculate is because they "rapidly differentiate into mature chromaffin cells." Quantitative data would enable comparison of data in Figure 6 in MYCN overexpressed *nf1* wt and mutant backgrounds to baseline levels in Figure 6—figure supplement 1.

However, the second and third points are arguably beyond the scope of the current paper and we would not require these to be addressed unless the authors wish to do so. These points are:

2) Figure 4, if they are aiming to test dose dependence in loss of NF1 alleles, why not go a step further with *nf1a -/- nf1b +/-*, since this would probably correlate with the data they showed in 2C?

3) Figure 5 suggests that NF1 loss blocks apoptosis that is otherwise induced in response to MYCN. Can the authors go further here, to address specific pathways downstream of RAS are blocking apoptosis? P38 would be the obvious candidate here.

*Reviewer #1:*

He and colleagues present a comprehensive and well conducted study of the collaboration between MYCN amplification and loss of the *nf1* tumour suppressor gene in a zebrafish model of neuroblastoma. The authors show that *nf1* loss potentiates transformation by MYCN-overexpression in zebrafish neuroblastoma. *nf1* deficiency in their model caused aberrant activation of RAS-MAPK signalling in MYCN-induced neuroblastoma, which promoted both tumour cell survival and proliferation, accelerating tumour onset and penetrance. Finally, using a series of in vivo structure-function analyses they show that the tumor suppression function of NF1 is mediated through the GAP activity of its GRD domain. Interesting differences in the capacity of this signal mechanism to suppress PSNS overgrowth in *nf1* deficient fish during development, further support the notion that a different and unknown activity of NF1 restricts the normal growth of PSNS neuroblasts during embryogenesis. This is a well done, interesting, important and informative study. The experiments are largely well conducted, the results clearly presented, and the data support the conclusions of the authors. While there are some minor issues with the experiments and additional work could be suggested, it is unlikely that these would alter what is an already excellent and interesting paper.

*Reviewer #2:*

In a transgenic zebrafish model of neuroblastoma, He and colleagues demonstrate that loss of *nf1a* coupled with MYCN overexpression accelerates neuroblastoma tumorigenesis and that this is due to activation of RAS signaling as this phenotype is suppressed with expression of the NF1 GRD domain. Additionally, they find synergy between trametinib and isotretinoin treatment for the MYCN overexpression, *nf1a* deficient neuroblastoma tumors, and suggest that their zebrafish model for aggressive neuroblastoma can be useful for studying additional therapeutics.

Overall, the study is rigorous, the reasoning and experiments are presented in a straightforward way, and all of the author's data and conclusions are sound.

Questions and concerns:

How do levels of *nf1b* compare to *nf1a* after the first week of zebrafish development? Do the authors have any insights into how loss of *nf1a* drives proliferation here?

Figure 4, if they are aiming to test dose dependence in loss of NF1 alleles, why not go a step further with *nf1a -/- nf1b +/-*, since this would probably correlate with the data they showed in 2C?

Figure 5 suggests that NF1 loss blocks apoptosis that is otherwise induced in response to MYCN. Can the authors go further here, to address specific pathways downstream of RAS are blocking apoptosis? P38 would be the obvious candidate here.

In Figure 6 and Figure 6—figure supplement 1, can data be quantified, especially given the result that *nf1a* loss does not seemingly affect RAS downstream signaling in non-MYCN amplified cells = no effect on endogenous levels in normal sympathoadrenal cells which the authors speculate is because they "rapidly differentiate into mature chromaffin cells." Quantitative data would enable comparison of data in Figure 6 in MYCN overexpressed *nf1* wt and mutant backgrounds to baseline levels in Figure 6—figure supplement 1.

---

## [Author Response]

Essential revisions:

Reviewer #2 presents four very interesting and pertinent points that could significantly improve the quality of the manuscript. Having reviewed these carefully, we suggest that the authors respond to the first and last of these four points, specifically:

1) How do levels of nf1b compare to nf1a after the first week of zebrafish development?

We have examined the expression of both *nf1a* and *nf1b* after the first week of zebrafish development (3, 4 and 6 weeks post fertilization) by qRT-PCR. We were only able to obtain qRT-PCR data for the whole animal at these time, because the Cherry labeled cells in the IRG and perpheral ganglia became a much lower percentage of the total cells and were difficult to dissociate in the juvenile fish. We attempetd to sort the <.1% of cells that were Cherry-positive, but the RNA from sorted cells was degraded.

In RNA extracted from the whole juvenile fish, we found that *nf1a* and *nf1b* levels were quantitatively similar at 3 to 6 wpf, and that both were decreased in relative terms compared to whole embros at 1 wpf.

In the revised manuscript, this point is clarified in the Results section as follows:

“Homozygous loss of *nf1a* led to the same level of increase in SCG neuronal cell number as homozygous loss of *nf1a* plus *nf1b*, while the loss of *nf1b* had little effect on SCG cell numbers (Figure 1), which is consistent with the fact that *nf1a* is expressed at a much higher level than *nf1b* in sympathetic neurons as well as the whole embryo during the first week of zebrafish embryonic development (Figure 1). Later in development at 3, 4 and 6 weeks of life, we observed lower relative *nf1a* and *nf1b* levels in RNA from the whole fish, and at these time points the expression levels of *nf1a* and *nf1b* were similar to each other, without evidence of the predominance of *nf1a* that was observed at 1 week of age.”

We also modified the Figure 1 to include these data.

We also inserted a sentence into the Materials and methods as follows: “For juvenlie zebrafish at the ages of 3, 4, and 6 weeks, 5 wild-type fish at each age were homogenized with a tissue grinder for RNA extraction.”

Do the authors have any insights into how loss of nf1a drives proliferation here?

In an attempt to address this point, we isolated RNAs from the tumors of *nf1a+/+;nf1b+/+;MYCN* and *nf1a-/-;nf1b+/+;MYCN* zebrafish and perormed qRT-PCR to assess expression of key genes involved in control of the cell cycle. We tested *ccna1, ccna2, ccnb1, ccnd1, ccnd2, ccnd3, ccne, cdk2, cdk4, cdk6 and e2f1*. However, we did not observe significant differences in RNA expression levels of these genes between the tumors with these two genotypes. Because the proteins encoded by these genes are regulated post-transcriptionally, their enzymatic activity will not be reflected in the RNA levels. Thus, the RNA levels alone are not enough to elucidate the mechanism driving increased proliferation. Unfortunately, antibodies for biochemistry are still lacking in the zebrafish, beyond the PCNA immunohistochemistry that we have done to demonstrate an increased fraction of cycling cells in the *nf1a* mutant tumors.

In the revised manuscript, we modified the Results section and added a supplementary figure (Figure 5—figure supplement 1) to include these data.

We modified the Results section as follows:

“We then isolated established neuroblastoma tumors of *nf1a+/+;nf1b+/+;MYCN* and *nf1a-/-;nf1b+/+;MYCN* zebrafish to examine expression of key genes involved in control of the cell cycle, including *ccna1, ccna2, ccnb1, ccnd1, ccnd2, ccnd3, ccne, cdk2, cdk4, cdk6 and e2f1*. We did not detect significant differences in mRNA levels of these genes (Figure 5—figure supplement 1), suggesting that further experiments are required to decipher the molecular mechanism through which the loss of nf1a promotes increased proliferation of MYCN-overexpressing neuroblastoma tumor cells.”

We have added the primer sequences into the Materials and methods as follows:

“*ccna1* forward, 5’- TGGCTCAGGGTCATTTATGG-3’ and reverse, 5’- TAACTTCGCATTCACGCAGG-3’; *ccna2* forward, 5’-TCCACTGGAGGCCAGTTTTG-3’ and reverse, 5’- GACTTGACCTCCATTTCCCG-3’; *ccnb1* forward, 5’-ATTCTCCTCAGTGTTTCTCCAGTC-3’ and reverse, 5’-AAGTGTAGATGTCTCGCTCATATTC-3’; *ccnd1* forward, 5’-TTGCTGCGAAGTGGATACCATAAG-3’ and reverse, 5’-AGGCACAATTTCTTTCTGAACACAC-3’; *ccnd2* forward, 5’- CCGTCCTGATCCGAATCTTCTG-3’ and reverse, 5’-GCCACCATCCTCCGCATAAAG-3’; *ccnd3* forward, 5’-ACGGCTACAGAGCTGAAGTT-3’ and reverse, 5’-CATCTGCTCGGCGCTAACA-3’; *ccne* forward, 5’-ACAACCTGCTCGGAAAAGACAAG-3’ and reverse, 5’-CACAAACCTCCATTAGCCAGTCC-3’; *cdk2* forward, 5’-TCGCGCTGAAGAAAATCCGA-3’ and reverse, 5’-ACGCAACTTGACTATGTTAGGGT-3’; *cdk4* forward, 5’-TGAGCCAGTAGCAGAGATCG-3’ and reverse, 5’-AGTGGGAGTCCGTCCTGATT-3’; *cdk6* forward, 5’-TCTCACCGTGTGGTTCATCG-3’ and reverse, 5’-ATGTCACAACCACCACGGAA-3’; *e2f1* forward, 5’-ACAACATCCAGTGGCTAGGG-3’ and reverse, 5’-TTCGTCCAGTTTCTCCTCGG-3’.”

4) In Figure 6 and Figure 6—figure supplement 1, can data be quantified, especially given the result that nf1a loss does not seemingly affect RAS downstream signaling in non-MYCN amplified cells = no effect on endogenous levels in normal sympathoadrenal cells which the authors speculate is because they "rapidly differentiate into mature chromaffin cells." Quantitative data would enable comparison of data in Figure 6 in MYCN overexpressed nf1 wt and mutant backgrounds to baseline levels in Figure 6—figure supplement 1.

As suggested by the reviewer #2, we have quantified the immunohistochemitry staining of the tumors (Figure 6) and IRG (Figure 6—figure supplement 1) to enable comparison of data in MYCN-overexpressing tumor cells (Figure 6) and normal sympathoadrenal cells without MYCN-overexpression (Figure 6—figure supplement 1). The quantified data is in agreement with our statement that elevated activation of RAS-MAPK effector pathways are only found in proliferating neuroblasts that were blocked from terminal differentiation by overexpression of MYCN.

In the revised manuscript, these data is added in the Results section as follows:

“The IRG of wild-type *nf1a+/+;nf1b+/+;EGFP* fish did not exhibit detectable phosphorylated ERK (pERK), phosphorylated AKT (pAKT) or phosphorylated S6 (pS6), demonstrating a very low level of basal RAS activity in the normal sympathoadrenal cells of the IRG (Figure 6—figure supplement 1). […] Thus, the loss of *nf1* did not cause a detectable increase in the basal levels of endogenous RAS signaling in normal sympathoadrenal cells of the IRG, probably because in the absence of MYCN overexpression they rapidly differentiate into mature chromaffin cells (Zhu et al. 2012).

We then assessed the role of *nf1* in suppressing RAS effector pathways in the neuroblastoma cells of wild-type fish overexpressing *MYCN*. By immunohistochemistry and measurement of the area of the diaminobenzidine-peroxidase (DAB)-stained tumor cells, we detected high levels of pERK, pAKT and pS6 in neuroblastomas from 6-week-old fish (Figure 6), indicating activation of RAS effector pathways in the proliferating neuroblasts of wild-type *nf1a+/+; nf1b+/+;MYCN* fishthat were blocked from terminal differentiation by overexpression of MYCN. In this context, it is not surprising that we observed much higher levels of activation of these effector pathways in neuroblastomas arising in *nf1a*-deficient fish, determined by significant increases of positive DAB-stained tumor areas (Figure 6). Thus, the overexpression of MYCN blocks the differentiation of sympathoadrenal precursor cells, leaving them vulnerable to the potentiating effects of *nf1* loss on RAS pathway activation, which in turn leads to the marked increased in neuroblastoma penetrance and growth rate.

In the revised manuscript, we added two figure panels to include the quantified data (Figure 6—figure supplement 1).

We also added the methods related to the quantification of IHC staining in the Materials and methods as follows:

“For quantification of immunohistochemistry staining, brightfield images were taken with the Mantra Quantitative Pathology Workstation (Perkin Elmer) and analyzed with ImageJ. The color deconvolution plugin of ImageJ and the “H DAB” vector was used to separate the hematoxylin and DAB stains (Ruifrok and Johnston, 2001), and the kidney tubules adjacent to the IRG or tumor region were applied as internal references to define threshold of DAB staining for each IRG or tumor region.”

However, the second and third points are arguably beyond the scope of the current paper and we would not require these to be addressed unless the authors wish to do so. These points are:

2) Figure 4, if they are aiming to test dose dependence in loss of NF1 alleles, why not go a step further with nf1a -/- nf1b +/-, since this would probably correlate with the data they showed in 2C?

We chose these representative genotypes in Figure 4 (*nf1a+/+; nf1b+/+, nf1a+/-;nf1b+/+* and *nf1a-/-;nf1b+/+*) for this analysis because nf1a is the dominant nf1 expressed in early development.

The *nf1a-/-; nf1b+/-; MYCN* fish are difficult to maintain because of the very early tumor onset and very high tumor penetrance (Figure 2), which means they often have to be sacrificed before they reach breeding age. We can only obtain the *nf1a-/-; nf1b+/-; MYCN* genotype by breeding *nf1a+/-; nf1b+/-; MYCN* fish with *nf1+/-;nf1b+/-* fish lacking MYCN transgene. Such breeding results in 16 viable genotypes, of which only one of eight MYCN transgenic fish have the *nf1a -/-;nf1b* +/- genotype.

3) Figure 5 suggests that NF1 loss blocks apoptosis that is otherwise induced in response to MYCN. Can the authors go further here, to address specific pathways downstream of RAS are blocking apoptosis? P38 would be the obvious candidate here.

This is a really good point for future studies involving specific inhibitors and genetic mutants of the ERK, SAPK/JNK and p38 MAPK pathways downstream of RAS signaling. We agree with the Reviewing Editor that this would be beyond the scope of the current paper, because these experiments would undoubtedly present difficulties in their execution and raise additional issues that will need to be addressed to have a complete story.